# Do Italian ObGyn Residents Have Enough Knowledge to Counsel Women About Nutritional Facts? Results of an On-Line Survey

**DOI:** 10.3390/nu17101654

**Published:** 2025-05-13

**Authors:** Laura Sarno, Dario Colacurci, Eleonora Ranieri, Rossella E. Nappi, Maurizio Guida

**Affiliations:** 1Department of Neurosciences, Reproductive Science and Dentistry, University Federico II, 80131 Naples, Italy; dario.colacurci@unina.it (D.C.); eleonoranieri@hotmail.it (E.R.); mauguida@unina.it (M.G.); 2Department of Clinical, Surgical, Diagnostic and Pediatric Sciences, University of Pavia, 27100 Pavia, Italy; nappi@rossellanappi.com; 3Research Center for Reproductive Medicine, Gynecological Endocrinology and Menopause, IRCCS S. Matteo Foundation, 27100 Pavia, Italy

**Keywords:** residents, specialty program, nutrition counseling, education, obstetrics, gynecology, female nutrition

## Abstract

**Background/Objectives**: Nutrition plays a key role in gynecologic conditions, pregnancy, and perinatal outcomes; therefore, nutritional counseling is essential in obstetric and gynecologic care. The present study aimed to delineate Italian Obstetrics and Gynecology (ObGyn) residents’ awareness of women’s nutrition and supplementation in different stages of life. **Methods**: We conducted a cross-sectional online survey about women’s nutrition and supplementation use throughout their lifetime. A 31-item survey questionnaire was distributed to Italian ObGyn Residents. **Results**: 258 Italian ObGyn Residents completed the online survey. About 40% of the responders considered their knowledge of women’s nutritional needs poor or very poor. A total of 226 residents (88%) declared that there is not enough time dedicated to women’s nutrition during their specialty program, and almost all the trainees would consider training in this area helpful for achieving a better professional profile. A total of 128 participants (49.6%) demonstrated insufficient knowledge in this field. Most (97.1%) of the responding trainees recommend supplementation during different stages of women’s lives. **Conclusions**: Italian ObGyn residents are not very skilled in women’s nutrition. There is an urgent need to develop specific training and interventions to educate our ObGyn residents about the importance of improving nutritional habits in patient care.

## 1. Introduction

Nutrition has been stated for decades as an essential promoter of health [1].

Growing evidence shows that maternal diet in pregnancy, breastfeeding, and early life nutrition can drive epigenetic changes, influencing susceptibility to several diseases in adulthood [2,3,4,5,6]. Both excessive and deficient intake of macro- and micronutrients can negatively affect pregnancy outcome and fetal development [7]. During pregnancy, the need for micronutrients increases more than macronutrients, placing the pregnant patient at a particularly high risk of deficiency, with significant consequences on maternal and perinatal outcomes. For example, folic acid deficiency can lead to neural tube defects, and iron deficiency is a known risk factor for maternal anemia and an increased risk of postpartum hemorrhage [8]. On one hand, maternal undernutrition is associated with growth restriction and fetal defects [9]. On the other hand, excessive nutrition can lead to macrosomia, childhood metabolic disease, obesity, type 2 diabetes mellitus, cardiovascular diseases [10], as well as pregnancy and delivery complications [11,12].

Apart from pregnancy, suboptimal eating habits might contribute to chronic and abnormal inflammatory responses, speeding up cellular senescence, with severe negative impacts on women’s health [13,14].

National and international guidelines and recommendations emphasize the importance of an accurate diet throughout different stages of female life [15,16,17]. However, they do not detail the significant hormonal changes occurring across aging. During fertile life, from menarche through peri- and postmenopausal period, it has become clear that the female body faces numerous endocrine fluctuations, with general and specific needs of nutrients [18]. Therefore, women’s diet composition should be modified according to women’s requirements. Due to a growing awareness of the importance of diet, we have been observing a widespread use of numerous supplements to support well-being both in physiological conditions and in case of obstetrics (e.g., conditions with an increased risk of preterm birth or hypertensive disorders of pregnancy), [19,20] or gynecologic problems (e.g., premenstrual syndrome [21], menstrual cycle disorders [22], or symptoms related to menopause [23]). However, their use should be tailored to the specific patient’s needs.

In this scenario, women’s nutrition represents an essential aspect in women’s care and should not be overlooked.

With this evidence, some questions arise: Are healthcare providers adequately informed about women’s nutritional needs to provide appropriate counseling? Do future Obstetricians and Gynecologists (ObGyns) training programs teach this information? What is young doctors’ awareness of the specific requirements of micronutrients in different stages of women’s lives? Is their prescription of supplements appropriate for women’s needs?

To address these points, we conducted a comprehensive nationwide study on behalf of the Italian Association of University Gynecologists (AGUI) to delineate the awareness of Italian Obstetrics and Gynecology (ObGyn) Residents regarding women’s nutrition and supplementation in different stages of life.

## 2. Materials and Methods

### 2.1. Study Design and Population

This was a cross-sectional online survey about women’s nutrition and supplementation use throughout their lifetime. The survey was promoted by the AGUI, an association joined by Professors and Researchers in Obstetrics and Gynecology who work in Italian University Hospitals and are involved in medical students’ and residents’ training. It was an online survey, disseminated among all the University Specialty Programs in Obstetrics and Gynecology in the country, from October to December 2024. In Italy, the Specialty Program in Obstetrics and Gynecology is a 5-year training program attended by medical doctors who want to become specialists in Obstetrics and Gynecology. Only Italian ObGyn Residents were eligible to participate in the survey. Participation in the survey was completely voluntary.

### 2.2. Survey Questionnaire

The research team developed an anonymous, self-completed, and structured questionnaire, including 31 items that could be completed in around 10 min.

The survey included four different parts:(1)Participants’ characteristics: age, gender, region of practice, year of residency, smoking, alcohol consumption, physical activity, supplementation use, and having children.(2)Personal perceptions about the importance of the current topic: how much do they feel confident about women’s nutrition, and if they think this topic deserves more time and space in the Specialty Program.(3)Residents’ knowledge: participants were asked to answer 9 simple questions about:
-recommended percentage of macronutrients (proteins, lipids, and carbohydrates) intake during a balanced meal [24];-recommended portions of fruit and vegetables per day [24];-recommended portions of fish per week [24];-types of food rich in Vitamin D [17];-recommended sugar intake per day [24];-role of dried fruit in pregnancy [25];-Vitamin D recommended intake in menopause [26];-nutritional advice in premenstrual syndrome [27];-nutritional advice for obese pregnant women [8].

These questions were elaborated according to current Italian and international evidence and recommendations [8,17,24,26,27,28], and they aimed to assess ObGyn trainees’ depth of knowledge about diet aspects in different stages of women’s lives.

Participants’ knowledge was considered sufficient when they correctly answered at least 6 out of 9 questions (2/3 of the total questions).

(4)Prescription of Supplements by Italian residents in Obstetrics and Gynecology: questions regarding when and which type of supplements they generally recommend.

### 2.3. Statistical Analysis

Results were presented as proportions of respondents to individual questions.

The sample was described by means of the usual descriptive statistics: mean and standard deviation for quantitative variables, while frequencies and percentages were used to describe the categorical variables. Chi-squared and Fisher’s exact tests were used to compare categorical variables between groups, while Student’s *t*-test and Mann–Whitney U test were used for continuous variables, and a parametric analysis of variance (ANOVA) was used to determine the differences when participants belonged to more than two groups. *p*-values < 0.05 were deemed statistically significant. All statistical analyses were carried out using SPSS statistics 29.0 (IBM Corporation, Armonk, NY, USA).

## 3. Results

### 3.1. Participant’s Characteristics

258 Italian trainees in Obstetrics and Gynecology completed the online survey during the study period. The main participants’ characteristics have been summarized in Table 1. There was a predominance of female against male and not declared gender participants (76.7% vs. 22.9% vs. 1%). The majority of responding trainees were from Southern and Islands Italy (49.2%), but there was a fair contribution also from Northern and Central Italy. Almost 70% of trainees were not smokers, and consumed alcohol once a week or less. Of note, less than one third met the recommended standard for routine physical activity [29].

### 3.2. Personal Perceptions About the Importance of the Current Topic

According to their personal perception, Italian trainees rated their knowledge regarding women’s nutritional needs at various stage of life as excellent (*n* = 3; 1.2%); very good (*n* = 42; 16.3%); good (*n* = 30; 11.6%); adequate (*n* = 81; 31.4%); poor (*n* = 85; 32.9%); and very poor (*n* = 17; 6.6%).

A total of 226 residents (88%) declared that there is not enough time for learning women’s nutrition during their specialty program, and 225 trainees (98.8%) would consider a training on women’s nutrition useful for their cultural background of future ObGyns. Only 42 participants (16.3%) reported that they work in hospitals with a clinic dedicated to women’s nutrition. The percentage of hospitals with a dedicated clinic was significantly higher in Northern Italy, compared to Southern and Islands and Central Italy (47.6% vs. 26.2% vs. 26.2%, respectively; *p* = 0.004).

### 3.3. Residents’ Knowledge

As described above, participants were asked to answer nine simple questions on different topics about women’s nutrition.

The percentage of correct answers for each question is reported in Table 2.

Only one resident (0.4%) answered all nine proposed questions correctly. 130 participants (50.4%) reported at least six correct answers and were classified as “residents with sufficient knowledge”.

Among respondents, those who were smokers and practicing in Southern Italy and the Islands were significantly more frequently participants without sufficient knowledge (smokers 26.6% vs. non-smokers 19.2%, *p* = 0.032; Southern and Islands Italy 57.8% vs. Northern Italy 21.9% vs. Central Italy 20.3%; *p* = 0.008).

### 3.4. Prescription of Supplements by Italian Residents in Obstetrics and Gynecology

A total of 212 ObGyn residents (82.2%) stated that patients’ eating habits cannot ensure an adequate intake of macro and micronutrients, and 245 (95%) participants declared that nutritional needs could differ according to patients’ cultural background.

The vast majority (97.1%) of the responding trainees recommend supplementation during different stages of women’s life (pregnancy, breastfeeding, preconception, menopause) or under specific circumstances (premenstrual syndrome, menstrual disorders, hormonal contraception). Only 5% of participants recommend supplementation in all these conditions (Figure 1).

According to the survey, the main reasons for not recommending supplementation could be the high costs (76.1%) and mistrust of effectiveness (51.8%). As a possible reason, the fear of side effects was almost not considered (0.7%), while 32.9% of participants reported that there is little information about the benefits.

Figure 2 reports the agreement rate about routine supplementation with common micronutrients (vitamin D, docosahexaenoic acid (DHA), and magnesium) in different conditions.

Regarding Vitamin D, 90.7% of participants considered routine supplementation useful during menopause. In contrast, routine supplementation of magnesium was considered beneficial mostly in premenstrual syndrome (71.7%) and pregnancy (69%), and routine DHA supplementation was appropriate in pregnancy by 60.9% of participants.

Finally, considering iron and folate as the two most common supplements, we asked which iron and folate supplements they generally recommend, when necessary; the answers are summarized in Table 3.

## 4. Discussion

Nowadays, a healthy lifestyle, based on regular physical activity and good eating habits, is an essential contributor to general well-being and a known protective factor for non-communicable diseases, such as metabolic diseases and tumors [29,30]. There is growing evidence that nutrition plays a significant role in developing gynecologic conditions [31], pregnancy outcomes, and short- and long-term neonatal outcomes [14].

Therefore, as specialists in obstetrics and gynecology, we can no longer ignore the importance of women’s nutrition in patient care.

With this cross-sectional study, we aimed to delineate the awareness of Italian ObGyn residents regarding women’s nutrition and supplementation in different stages of life, to implement our teaching offer according to the results. Firstly, we found that around 1 in 3 residents considered their knowledge of women’s nutritional needs poor or very poor. Moreover, most participants declared that there is not enough time dedicated to women’s nutrition during their specialty program, and almost all the trainees would consider training in this area helpful in achieving a better professional profile. When we addressed residents’ knowledge, we realized that only 1 in 2 participants could answer at least 2/3 of the questions correctly. Those with less competence were more likely to be smokers, thus they displayed an unhealthy habit, leading to the hypothesis that trainees with a less healthy lifestyle are also less keen to consider nutrition and lifestyle advice as an essential part of women’s counseling and management. Moreover, they were more likely to work in Southern Italy and the Islands, a feature consistent with the fact that most of the hospitals in these regions did not have a dedicated clinic. In general, the percentage of participants reporting to work in a hospital with a clinic specialized in women’s nutrition was very low (16.3%), highlighting how the importance of nutritional counseling to support women with gynecologic problems or during specific life stages, such as pregnancy and post-menopause, is still underestimated. This gap in residents’ education is consistent with the results by Hachey et al., addressing nutrition education and nutrition knowledge among ObGyn residents across the United States in 2022 [32]. Indeed, almost half of the participants reported 0 h per year of dedicated education on this topic, and less than one-third of trainees felt confident counseling patients on nutrition in pregnancy [21]. Moreover, as reported in our study, they realized there is a disconnection between the recognized importance of nutrition and the lack of residents’ knowledge [32].

Most participating residents (82.2%) considered that patients’ eating habits cannot ensure an adequate intake of micro- and macro-nutrients. This could be in line with the results of an Italian multicenter prospective cohort study recruiting Italian healthy normal-weight women, reporting that micronutrient intakes were below the recommended range in this population [33].

According to our survey, prescribing supplements is a prevalent practice among our residents, especially during pregnancy and breastfeeding. (Figure 1) However, when we deeply interrogated when and how we should prescribe supplements, answers were very heterogeneous. We therefore concluded there is a lack of a common direction on which supplements are genuinely beneficial. The use of multivitamins in pregnancy has been widely spread in different settings [34]. However, an excessive intake of some micronutrients might lead to unexpected and undesired effects. Indeed, some authors reported that high doses of folic acid might increase circulating unmetabolized folic acid, perturbing one-carbon metabolism and leading to detrimental effects, such as pseudo-MTHFR deficiency and vitamin B12 deficiency [35,36]. Moreover, iron overload has been associated with increased oxidative stress that can be involved in some obstetric complications, such as preeclampsia [37]. Starting from this evidence, it becomes mandatory to sensitize our residents to the importance of improving dietary adequacy among our patients and of using supplementation as a helpful tool, only when it is appropriate [38].

Historically, nutritional counseling has been almost entirely assigned to nutritionists and dietitians. Being supported by these professionals is fundamental in our field, considering that many conditions are directly linked to nutrition, and ObGyns might not be skilled enough to support their patients alone. Moreover, ObGyns have to face the short time available for patients’ examination, making it somewhat difficult to save sufficient time to discuss nutrition. If, on the one hand, the short duration of clinical examination poses an essential limitation for integrating nutritional counseling into routine patients’ care, on the other hand, there is a shortage of nutritionists and dietitians in many multidisciplinary teams, as confirmed by our findings. A significant challenge would certainly be improving the presence of a dedicated nutritionist or dietitian in an ObGyn setting. However, in this healthcare reality, it is essential to implement ObGyns’ nutrition knowledge during their training to provide basic and evidence-based guidance to their patients.

According to the Italian Ministry of Health (D.I. n. 402/2017) [39], the curriculum of the residency program in Gynecology and Obstetrics does not mandate specific hours of lectures dedicated to nutrition. Each school can decide whether to include the subject. Our data could incentivize Italian Specialty Programs in Obstetrics and Gynecology to increase the hours devoted to women’s nutrition.

The main strength of our study is that, to our knowledge, this is the first Italian study assessing ObGyn residents’ insight into women’s nutrition needs throughout their lifetime. In addition, our cross-sectional survey involved trainees from several Italian regions, giving a picture of the status across the country. However, the study has potential limitations. Firstly, surveys are subject to selection bias, and it might have happened that participants were those residents who were more interested in the topic. Secondly, we did not perform a formal validation of the survey instrument. Moreover, the sample size is relatively small, considering that the population of ObGyn residents in Italy is around 2000. Also, the low percentage of participation in the survey might be interpreted as a confirmation of a lack of interest, along with an underestimation of the importance of this topic by our residents. However, to our knowledge, the only other study available on this topic was performed in the United States and involved an even smaller sample size of 218 participants [32].

## 5. Conclusions

According to the present Italian cross-sectional survey, residents are not sufficiently skilled in women’s nutrition throughout their lifetime, and their choice regarding supplements is quite heterogeneous. Therefore, their current knowledge should be improved to avoid malpractice. We strongly believe there is an urgent need to develop specific training and interventions to educate our ObGyn residents about the importance of enhancing nutritional habits in patient care.

## Figures and Tables

**Figure 1 nutrients-17-01654-f001:**
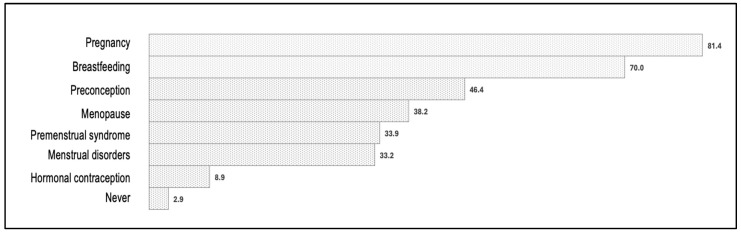
Percentage of ObGyn trainees recommending supplementation during pregnancy, breastfeeding, preconception, menopause, premenstrual syndrome, menstrual disorders, and hormonal contraception.

**Figure 2 nutrients-17-01654-f002:**
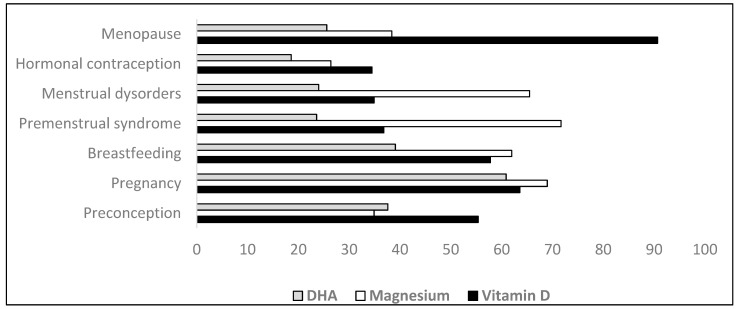
Participants’ agreement rate regarding routine supplementation with Vitamin D, Magnesium, and DHA in different women’s conditions.

**Table 1 nutrients-17-01654-t001:** Main participants’ characteristics.

Characteristics	*n* = 258 (%)
**Gender**	
Male	59 (22.9)
Female	198 (76.7)
Not declared	1 (0.4)
**Age**	29.4 (2.6)
**Region of practice**	
Northern Italy	78 (30.2)
Central Italy	53 (20.5)
Southern and Islands Italy	137 (49.2)
**Year of Residency**	
1st	60 (23.3)
2nd	34 (13.2)
3rd	66 (25.6)
4th	71 (27.5)
5th	25 (9.7)
Unknown	2 (0.8)
**Smoking**	
Yes	59 (22.9)
No	179 (69.4)
Ex smoker	16 (6.2)
Unknown	4 (1.6)
**Alcohol consumption**	
>3 times/week	3 (1.2)
2–3 times a week	26 (10.1)
Once a week or less	172 (66.7)
No	53 (20.5)
unknown	4 (1.6)
**Physical activity**	
<150 min/week	191 (74.0)
≥150 min/week	67 (26.0)
**Use of supplements**	
yes	99 (38.4)
**Having children**	
yes	13 (5.0)

**Table 2 nutrients-17-01654-t002:** Percentage of correct answers by question.

Topics	Correct Answers *n* (%)
Macronutrient distribution	40 (15.5)
Portions of fruit and vegetables per day	106 (41.1)
Portions of fish per week	227 (88.0)
Food rich in vitamin D	129 (50)
Sugar intake	173 (67.1)
Dried fruit in pregnancy	200 (77.5)
Nutrition and menopause	210 (81.4)
Nutrition and premenstrual syndrome	155 (60.1)
Nutrition in obese pregnant women	178 (69)

**Table 3 nutrients-17-01654-t003:** Type of recommended iron and folate supplementation.

**Folate Supplementation**	**N (%)**
5-Methyl-tetrahydrofolate	77 (29.8)
Folic Acid	130 (50.4)
It’s the same	7 (2.7)
I don’t know the difference	44 (17.1)
**Iron Supplementation**	**N (%)**
Ferrous sulphate	180 (7.0)
Liposomal or sucrosomial iron	83 (32.2)
Iron bysglicinate	5 (1.9)
Ferrous sulphate as first line and other types as second line	97 (37.6)
It’s the same	16 (6.2)
I don’t know the difference	39 (15.1)

## Data Availability

The original contributions presented in this study are included in the article. Further inquiries can be directed to the corresponding author.

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
