# Peer review of "Do Italian ObGyn Residents Have Enough Knowledge to Counsel Women About Nutritional Facts? Results of an On-Line Survey"

_nutrients, 2025, doi:10.3390/nu17101654_

Round 1

Reviewer 1 Report

Comments and Suggestions for Authors

The topic addressed by the manuscript is highly relevant, current, and raises important discussions regarding medical training. The inclusion of nutrition-related content in residency programs in Gynecology and Obstetrics appears to be a necessary strategy, particularly in light of the growing demand for a more comprehensive approach to women's health.

Historically, nutritional counseling has been almost exclusively assigned to nutritionists. However, the reality within healthcare services—especially in primary care—reveals a shortage of these professionals in many multidisciplinary teams. Moreover, although nutritionists are the primary professionals in this field, physicians can and should provide basic, evidence-based guidance, particularly in routine situations such as prenatal care.

The proposal to incorporate nutrition content into the training of gynecology residents is both timely and necessary, considering that many clinical and obstetric conditions are directly linked to dietary and nutritional factors.

It is important to note, however, the limitation posed by the short duration of medical appointments, especially in prenatal care, where consultations are often restricted to brief periods (sometimes less than 15 minutes). This represents a significant challenge for integrating new responsibilities into clinical practice. Nonetheless, it is a challenge that must be addressed, given that the quality of care provided to pregnant women must be comprehensive, effective, and patient-centered.

In summary, the manuscript offers a valuable contribution to the reflection and enhancement of medical education by emphasizing the importance of integrating nutritional aspects into clinical gynecological and obstetric practice.

Author Response

  1. Summary

We thank the reviewer for the positive comments and have tried to improve the manuscript accordingly.

Please find the detailed responses below and the corresponding revisions/corrections highlighted in the re-submitted files.

2. Questions for General Evaluation

Reviewer’s Evaluation

Response and Revisions

Does the introduction provide sufficient background and include all relevant references?

yes

We thank the Reviewer for his/her positive comments

Is the research design appropriate?

yes

Are the methods adequately described?

yes

Are the results clearly presented?

yes

Are the conclusions supported by the results?

yes

  1. Point-by-Point response to Comments and Suggestions for Authors

Comments 1: The topic addressed by the manuscript is highly relevant, current, and raises important discussions regarding medical training. The inclusion of nutrition-related content in residency programs in Gynecology and Obstetrics appears to be a necessary strategy, particularly in light of the growing demand for a more comprehensive approach to women's health.

Historically, nutritional counseling has been almost exclusively assigned to nutritionists. However, the reality within healthcare services—especially in primary care—reveals a shortage of these professionals in many multidisciplinary teams. Moreover, although nutritionists are the primary professionals in this field, physicians can and should provide basic, evidence-based guidance, particularly in routine situations such as prenatal care.

The proposal to incorporate nutrition content into the training of gynecology residents is both timely and necessary, considering that many clinical and obstetric conditions are directly linked to dietary and nutritional factors.

It is important to note, however, the limitation posed by the short duration of medical appointments, especially in prenatal care, where consultations are often restricted to brief periods (sometimes less than 15 minutes). This represents a significant challenge for integrating new responsibilities into clinical practice. Nonetheless, it is a challenge that must be addressed, given that the quality of care provided to pregnant women must be comprehensive, effective, and patient-centered.

In summary, the manuscript offers a valuable contribution to the reflection and enhancement of medical education by emphasizing the importance of integrating nutritional aspects into clinical gynecological and obstetric practice.

Reply: We thank the Reviewer for this consideration, which could be very valuable to our manuscript. We highlighted this interesting point in the discussion (line 259-271) .

Reviewer 2 Report

Comments and Suggestions for Authors

Thank you for granting me to review the MS entitled "Do Italian ObGyn residents have enough knowledge to counsel women about nutritional facts? Results of an on-line survey". I think that the MS is on an important knowledge gap of the obstetricians resistence on necessary micro and macronutrients during pregnancy. 

I have some minor remarks:

The introduction is very diffuse and not focusing on the concrete needs on nutrients during pregnancy. There are many general sentences why is important to take nutrients and what are the consequences of the incomplete supply of nutrients. There are some lines that are beyond the scope (between lines 46-59). The sentences between lines 54 and 57 are practically repetitions. The sentences between lines 61 and 67 are also beyond the scope. I recommend to rewrite the Introductory section with more concrete physiological facts.

The shortening AGUI is not necessary to be written out in the line 82-83, because it is explained in the text earlier before.

The recommended standards regarding the nutrients should be interpreted in a more factual detailed ways between lines 97 and 103. The interpretation of the questions in the questionnaires should be more elaborated in the Materials and Methods section.

Figure 1 is lengthy and not necessary. In my opinion, Figure 2 should be also ommitted.

Results between lines 164 and 166 should be condensed.

The Discussion is relatively short as compared to the Results section. It is not so factual and there are some straight repetitions of the results (between lines 213 and 215). 

Author Response

Reviewer 2:

  1. Summary

We thank the reviewer for the valuable comments and have tried to improve the manuscript accordingly.

Please find the detailed responses below and the corresponding revisions/corrections highlighted in the re-submitted files.

2. Questions for General Evaluation

Reviewer’s Evaluation

Response and Revisions

Does the introduction provide sufficient background and include all relevant references?

Can be improved

We thank the Reviewer for his/her comments and we hope to have implemented all the required changes in such a manner that makes the manuscript now suitable for publication

Is the research design appropriate?

Yes

Are the methods adequately described?

Yes

Are the results clearly presented?

Can be improved

Are the conclusions supported by the results?

Yes

  1. Point-by-Point response to Comments and Suggestions for Authors

Comments 1: The introduction is very diffuse and not focusing on the concrete needs on nutrients during pregnancy. There are many general sentences why is important to take nutrients and what are the consequences of the incomplete supply of nutrients. There are some lines that are beyond the scope (between lines 46-59). The sentences between lines 54 and 57 are practically repetitions. The sentences between lines 61 and 67 are also beyond the scope. I recommend to rewrite the Introductory section with more concrete physiological facts.

Reply: We thank the reviewer for the comment. However, it is important to point out that our manuscript aimed to address ObGyn trainees’ knowledge about women’s nutrition through life (from puberty to menopause), and it did not refer only to nutrition in pregnancy. This can partially explain why the reviewer thinks that some lines are beyond the scope, while we  think that some observations are crucial in the introduction. (eg “Through the menarche, during fertile life, through peri-and postmenopausal period, it has become clear that women body faces numerous endocrine fluctuations, with general and specific need of nutrients. [17] Therefore, women’s diet composition should be modified according to women’s requirements. In this scenario, women’s nutrition represents an important aspect in women’s care, and it should not be overlooked”)

Nevertheless, we agree that the introduction should be more focused, and we have tried to remove repetitions and to improve it.  

Comment 2: The shortening AGUI is not necessary to be written out in the line 82-83, because it is explained in the text earlier before.

Reply: We thank the Reviewer for the kind remark and have corrected accordingly. (line. 80)

Comment 3: The recommended standards regarding the nutrients should be interpreted in a more factual detailed ways between lines 97 and 103. The interpretation of the questionnaire questions should be more elaborated in the Materials and Methods section.

Reply. We thank the reviewer for the comment. We reported more details regarding the questionnaire and its interpretation. (line 100-117)

Comment 4: Figure 1 is lengthy and not necessary. In my opinion, Figure 2 should be also ommitted.

Reply: We thank the Reviewer for the comment. We have removed the figures and modified the manuscript accordingly.

Comment 5: Results between lines 164 and 166 should be condensed.

Reply: We thank the reviewer for the valuable comment and have tried to improve the manuscript accordingly. (line 145-148)

Comment 6: The Discussion is relatively short as compared to the Results section. It is not so factual and there are some straight repetitions of the results (between lines 213 and 215).

Reply: We thank the Reviewer for the comment. We have tried to improve and extend the discussion and have also removed straight repetitions of the results. 

Reviewer 3 Report

Comments and Suggestions for Authors

The manuscript involves an important topic whether obstetricians and gynecologists have proper nutrition knowledge to advise their female patients. However, some amendments must be made to improve the overall quality of the manuscript.

Please pay attention to proper English, as some phrases seem to be a bit awkward, for example “defective intake” (wouldn’t “deficient intake” sound better?) or “growth restriction” (maybe “growth impairment”?)

-Line 53 – what kind of guidelines do you mean here? The Italian ones?

-Did the study receive the approval of the Bioethics Committee? If so, provide the approval number.

-More details regarding inclusion and exclusion criteria are needed. As you wrote, the questionnaire was disseminated among the members of the Italian Association of University Gynecologists. Are all of its members practicing at hospitals/outpatient clinics?

-It must also be highlighted in Materials and Methods that only obstetricians and gynecologists were eligible to participate in the study.

-The Authors have already indicated the limitations of the study. However, the issue of not using a validated questionnaire may be considered another limitation.  

-Line 122 – “Two-hundred and fifty-eight Italian trainees in Obstetrics and Gynecology completed the online survey.” This issue must be clarified earlier in the Materials and Methods because in my country, obstetrics and gynecology are two different and separate degrees at the university and graduates have other types of knowledge (an obstetrician is not a doctor).

-Line 143 – please be consistent throughout the manuscript and use numbers instead of written descriptions of the number, so write 227, instead of two-hundred and twenty-seven. Apply it to the whole manuscript.

-Line 196 – what was the exact question regarding folate and iron supplementation? It matters because folate supplementation should be routine at the preconception stage and during pregnancy, but routine iron supplementation is not recommended if there is no anemia.  

-It would be interesting to include information on what kind of nutritional subjects students of obstetrics and gynecology have in their curriculum during their studies in the discussion. It would help to understand the obtained results (taking into account what you wrote – “88% of them declared that there is not enough time dedicated to women’s nutrition during their specialty program”)

-Line 230 – please cite more studies regarding ObGyn knowledge on nutrition. How does it look like in other European countries?

-Considering an inadequate knowledge of ObGyn, more study implications should be highlighted. For example, shouldn’t ObGyn advise their female patients to have a consultation with dietitians or nutritionists?

- Follow the journal's guidelines regarding references - the font, font size etc.

Author Response

  1. Summary

We thank the reviewer for the valuable comments and have tried to improve the manuscript accordingly.

Please find the detailed responses below and the corresponding revisions/corrections highlighted in the re-submitted files.

2. Questions for General Evaluation

Reviewer’s Evaluation

Response and Revisions

Does the introduction provide sufficient background and include all relevant references?

Can be improved

We thank the Reviewer for his/her comments and we hope to have implemented all the required changes in such a manner that makes the manuscript now suitable for publication

Is the research design appropriate?

Must be improved

Are the methods adequately described?

Must be improved

Are the results clearly presented?

Can be improved

Are the conclusions supported by the results?

Must be improved

  1. Point-by-Point response to Comments and Suggestions for Authors

Comments 1: Please pay attention to proper English, as some phrase seem to be a bit awkward, for example “defective intake” (wouldn’t “deficient intake” sound better?) or

“growth restriction” (maybe “growth impairment”?)

Reply: Thank you for your comment. We changed “defective intake” into “deficient intake”,

as suggested; however, with the term “growth restriction”, we referred to a specific complication of the pregnancy, characterized by a failure of the fetus in reaching its genetic growth potential, which is commonly identified as fetal growth restriction. Please, see below some examples:

-DOI: 10.1002/uog.22134

-https://www.rcog.org.uk/guidance/browse-all-guidance/green-top-guidelines/small-for-

gestational-age-fetus-and-a-growth-restricted-fetus-investigation-and-care-green-top-

guideline-no-31/

Moreover, we edited the whole manuscript in English

Comment 2: Line 53 – what kind of guidelines do you mean here? The Italian ones?

Reply: Thank you for pointing this out. We refer to both national and international guidelines and recommendations. We clarified this in the manuscript and reported more references.  (line52-53)

Comment 3: Did the study receive the approval of the Bioethics Committee? If so, provide the approval number.

Reply: Thank you for your question. This study did not require approval by the ethics committee as it was conducted anonymously and did not involve clinical data or patient information. The purpose of the study was solely to assess the quality of educational training among junior doctors. Participation was voluntary, and no personal or sensitive data were collected.  Informed consent was obtained from all subjects involved in the study.

An Informed Consent Statement has been added to the manuscript. (line 04-305)

Comment 4: More details regarding inclusion and exclusion criteria are needed. As you wrote, the questionnaire was disseminated among the members of the Italian Association of University Gynecologists. Are all of its members practicing at hospitals/outpatient clinics?

Reply: We thank the Reviewer for the question. The Italian Association of University Gynecologists comprises Professors and Researchers in Obstetrics and Gynecology who practice in Italian University Hospitals and routinely participate in medical students’ and residents’ training. We specified this in the manuscript (line 80-82). Moreover, the study has been promoted by the AGUI, but the survey was disseminated among Italian ObGyn residents (please see also Reply to Comment 5).

Comment 5: It must also be highlighted in Materials and Methods that only obstetricians and gynecologists were eligible to participate in the study.

Reply: We thank the Reviewer for the suggestion. Only Italian ObGyn Residents were eligible, and we highlighted it in Materials and Methods (line 82-84).

Comment 6: The Authors have already indicated the limitations of the study. However, the issue of not using a validated questionnaire may be considered another limitation.  

Reply: We acknowledge the reviewer's important point about the survey instrument’s lack of formal validation. We understand that formal validation would enhance the rigor of our findings, and we have included this limitation in the discussion.  (line 281-282)

Comment 7: -Line 122 – “Two-hundred and fifty-eight Italian trainees in Obstetrics and Gynecology completed the online survey.” This issue must be clarified earlier in the Materials and Methods because in my country, obstetrics and gynecology are two different and separate degrees at the university and graduates have other types of knowledge (an obstetrician is not a doctor).

Reply: We thank the Reviewer for the comment. In Italy, we have a School of Midwife, which is a 3-year degree for students who want to practice as midwives; the Specialty Program in Obstetrics and Gynecology is a 5-year training program for Medical Doctors who wish to become specialists in Obstetrics and Gynecology. In Italy, an “Obstetrician” is a medical doctor who has completed the Specialty Program in Obstetrics and Gynecology and works mainly in the obstetrics field. Generally, ObGyn manuscripts use the term “obstetricians” referring to doctors who are specialists in Obstetrics.

We clarified this point in the materials and methods. (line 84-88)

Comment 8: Line 143 – please be consistent throughout the manuscript and use numbers instead of written descriptions of the number, so write 227, instead of two-hundred and twenty-seven. Apply it to the whole manuscript.

Reply: We thank the Reviewer for the suggestion. We modified the manuscript accordingly.

Comment 9: Line 196 – what was the exact question regarding folate and iron supplementation? It matters because folate supplementation should be routine at the preconception stage and during pregnancy, but routine iron supplementation is not recommended if there is no anemia.  

Reply: We thank the Reviewer for the question. Section 4 of the survey included two questions regarding folate and iron supplementation (Prescription of Supplements by Italian Residents in Obstetrics and Gynecology: Questions regarding when and which type of supplements they generally recommend). The questions related to the type of iron or folate they prefer to prescribe when necessary.

Specifically:

When a folate supplementation is recommended, which type do you prescribe?

  1. Folic acid
  2. 5-metyltetrahydrofolate
  3. It is the same
  4. I do not know the difference

When iron supplementation is recommended, which type of iron do you prescribe?

  1. Ferrous sulfate
  2. Liposomal or sucrosomial iron
  3. Bisglycinate iron
  4. Ferrous sulfate as a first instance, and liposomal or sucrosomial iron as a second instance in case of intolerance
  5. It is the same
  6. I do not know the difference

Participants reported very heterogeneous answers, as discussed in the discussion.

We did not report the entire questions since each answer is reported in Table 3.

Comment 10: It would be interesting to include information on what kind of nutritional subjects students of obstetrics and gynecology have in their curriculum during their studies in the discussion. It would help to understand the obtained results (taking into account what you wrote – “88% of them declared that there is not enough time dedicated to women’s nutrition during their specialty program”)

Reply: We thank the Reviewer for the kind request. According to the Italian Ministry of Health, decree D.I. n. 402/2017, the curriculum of the residency program in Gynecology and Obstetrics does not mandate specific hours of lectures dedicated to nutrition. Each school can decide whether to include the subject. Our data could incentivize Italian Specialty Programs in Obstetrics and Gynecology to increase the hours devoted to nutrition. We highlighted this point in the discussion. (line 271-275)

Comment 11: Line 230 – please cite more studies regarding ObGyn knowledge on nutrition. How does it look like in other European countries?

Reply: We thank the Reviewer for the kind request. To the best of our knowledge, there is no other study regarding ObGyn residents’ knowledge, apart from the one we have already cited. However, if the Reviewer knows about different studies on this topic, we would happily include them in the manuscript.

Comment 12: Considering an inadequate knowledge of ObGyn, more study implications should be highlighted. For example, shouldn’t ObGyn advise their female patients to have a consultation with dietitians or nutritionists?

Reply: We thank the reviewer for pointing this out.

Historically, nutritional counseling has been almost entirely assigned to nutritionists and dietitians. Being supported by a dietitian or a nutritionist is fundamental in our field, considering that many conditions are directly linked to nutrition, and ObGyns might not be skilled enough to support their patients alone. Moreover, ObGyns have to face the short time available for patients’ examination, making it somewhat difficult to save sufficient time to discuss nutrition. If, on one hand, the short duration of clinical examination poses an essential limitation for integrating nutritional counseling into routine patients’ care, on the other hand, there is a shortage of nutritionists and dietitians in many multidisciplinary teams. We reported that a low percentage of hospitals have a dedicated clinic for women’s nutrition. A significant challenge would certainly be improving the presence of a dedicated nutritionist or dietitian in an ObGyn setting. However, in this healthcare reality, it is essential to implement ObGyns’ nutrition knowledge during their training, to enable them to provide basic, evidence-based guidance to their patients, particularly in routine situations such as prenatal care.

We discussed this point in the discussion. (line 258-270)

Comment 13:- Follow the journal's guidelines regarding references - the font, font size etc.

Reply: We thank the Reviewer for the suggestion and have modified references accordingly.

Round 2

Reviewer 3 Report

Comments and Suggestions for Authors

The Authors have responded to all my questions.